# A Survey on Deep Learning in COVID-19 Diagnosis

**DOI:** 10.3390/jimaging9010001

**Published:** 2022-12-20

**Authors:** Xue Han, Zuojin Hu, Shuihua Wang, Yudong Zhang

**Affiliations:** 1School of Mathematics and Information Science, Nanjing Normal University of Special Education, Nanjing 210038, China; 2School of Computing and Mathematical Sciences, University of Leicester, Leicester LE1 7RH, UK

**Keywords:** COVID-19, diagnosis, deep learning, convolutional neural networks, CT images, transfer learning, X-ray images, classification

## Abstract

According to the World Health Organization statistics, as of 25 October 2022, there have been 625,248,843 confirmed cases of COVID-19, including 65,622,281 deaths worldwide. The spread and severity of COVID-19 are alarming. The economy and life of countries worldwide have been greatly affected. The rapid and accurate diagnosis of COVID-19 directly affects the spread of the virus and the degree of harm. Currently, the classification of chest X-ray or CT images based on artificial intelligence is an important method for COVID-19 diagnosis. It can assist doctors in making judgments and reduce the misdiagnosis rate. The convolutional neural network (CNN) is very popular in computer vision applications, such as applied to biological image segmentation, traffic sign recognition, face recognition, and other fields. It is one of the most widely used machine learning methods. This paper mainly introduces the latest deep learning methods and techniques for diagnosing COVID-19 using chest X-ray or CT images based on the convolutional neural network. It reviews the technology of CNN at various stages, such as rectified linear units, batch normalization, data augmentation, dropout, and so on. Several well-performing network architectures are explained in detail, such as AlexNet, ResNet, DenseNet, VGG, GoogleNet, etc. We analyzed and discussed the existing CNN automatic COVID-19 diagnosis systems from sensitivity, accuracy, precision, specificity, and F1 score. The systems use chest X-ray or CT images as datasets. Overall, CNN has essential value in COVID-19 diagnosis. All of them have good performance in the existing experiments. If expanding the datasets, adding GPU acceleration and data preprocessing techniques, and expanding the types of medical images, the performance of CNN will be further improved. This paper wishes to make contributions to future research.

## 1. Introduction

WHO declared COVID-19 as a global pandemic in March 2020. The COVID-19 pandemic has affected thousands of people. It has affected people’s ordinary lives and the global economy. COVID-19 can cause respiratory, gastrointestinal, and neurological syndromes [1]. Cough, fever, and other respiratory issues are the most common symptoms [2].

Rapid COVID-19 diagnosis is essential during the pandemic. Currently, the commonly used diagnostic methods include the molecular assay, chest computed tomography (CT) scan combined with the evaluation of clinical symptoms [3], artificial intelligence (AI) methods [4], potential electrochemical (EC) biosensors [5], surface plasmon resonance (SPR)-based biosensors [6], field-effect transistor (FET)-based biosensors [7], etc. As a frequently employed auxiliary detection technology, CT images can show the changes in the lung caused by virus infection [8]. Compared with CT images, X-ray images are more accessible to obtain. It is also an important means of medical detection.

The most common diagnostic measure for COVID-19 is through reverse transcription-polymerase chain reaction (RT-PCR) assays of nasopharyngeal swabs [9]. However, the high false negative rate of RT-PCR [10] may affect the timely treatment of infected patients. The X-ray and CT can image the lungs of patients with COVID-19. Lung imaging can reveal the niduses’ spatial location and the infection’s extent. CT images have a fast turnaround and excellent sensitivity [11]. They can visualize the degree of infection in the lung. Based on the significant features of COVID-19 on X-ray or CT images, many researchers have used artificial intelligence and computer vision to classify X-ray or CT images. These images were classified into two categories: those without COVID-19 and those infected with COVID-19. Other researchers divided the images into healthy, infected with COVID-19, and infected with pneumonia. Some algorithms can detect the extent of infection based on the features of X-ray or CT images. This research is meant to help doctors diagnose COVID-19 accurately and quickly.

Artificial intelligence is widely used in medical [12,13,14]. Its accuracy rates and prediction are high [15]. AI can be applied to multiple phases, such as prediction, diagnosis, virus detection, response, prevention, and recovery, to accelerate research [16,17]. During the COVID-19 epidemic, AI recognized chest X-rays or CT images. Features in X-ray or CT images are extracted for COVID-19 diagnosis by segmenting regions of interest and capturing fine structures. One of the important subfields of AI is machine learning (ML) [18]. It is already widely applied to medical images [19,20,21]. Deep learning (DL) is a promising technology in machine learning [22]. Deep learning has multiple hidden layers for learning and can perform classification or detection tasks well [23,24]. The role of deep learning in image recognition is vital [19,25,26,27]. The convolutional neural network (CNN) is a kind of deep network. It is popular in computer vision applications. CNN has been successfully applied to biological image segmentation [28], traffic sign recognition [29,30,31], face recognition [32,33], and other fields. Later studies added rectified linear units, dropout, data augmentation, and other techniques to CNN. This decreased the error rate of deep learning for image classification tasks to less than 3% in 2016 [34] and exceeded human performance.

This paper mainly summarizes and discusses methods and experiments based on convolutional neural networks, which classify chest X-ray or CT images into infected and non-infected with COVID-19. All along, CNNs have been very widely used in medical images [35,36,37]. The severity and spread of COVID-19 around the world are alarming. CNN is used to extract features from chest X-rays or CT images. CNN, combined with other algorithms or architectures, divided the images into two categories: infected with COVID-19 and uninfected, or three categories: infected with COVID-19, infected with other pneumonia, and uninfected [38,39,40]. The COVID-19 diagnosis based on convolutional neural networks can assist doctors in making judgments quickly and accurately. We gave a detailed explanation of CNN, its existing technologies, and network types. The following contents also summarize, analyze and compare the automatic diagnosis systems established by scholars based on convolutional neural networks. Finally, we suggested future research on improving the performance of COVID-19 classification models.

The parts of this paper are organized as follows: Section 2 reviews the characteristics of medical images currently commonly used for COVID-19 diagnosis. Section 3 introduces convolutional neural networks (CNN) and current methods commonly used to improve CNN. Section 4 introduces several mature and well-performing convolutional neural networks. Section 5 analyzes and discusses the current experiments and methods of applying CNN to COVID-19 diagnosis. Section 6 summarizes the full text and provides suggestions for future research. To help understand more clearly, all abbreviations and full names in this paper are shown in Table 1.

## 2. Imaging Modalities for COVID-19 Diagnosis

Computed tomography (CT), published in 1972 [41], has become a widely used tool for diagnostic imaging. CT is a cross-sectional scan of a certain part of the human body one by one. Its advantages include clear images and fast scanning. CT is widely used to detect a variety of diseases. X-rays penetrate a person’s body and take an X-ray image. X-ray images are also an important basis for diagnosing diseases.

### 2.1. Chest Computed Tomography

CT takes images from different angles. One shot was taken for each rotation angle. Using a large number of projection images taken from different angles, we back-calculated a fault plane image by a mathematical algorithm. This is computed tomography. Many scholars have studied the features of COVID-19 on CT images. In the study of patients in Rome and Italy [8], ground glass opacity (GGO) was found in 100% of confirmed patients on CT images. Ground-glass opacity (GGO) means that the density will be slightly increased on high-resolution CT images, and the bronchovascular will still be visible. This sign is often the manifestation of early lung disease. Timely detection and diagnosis of GGO are important for clinical management. Multilobe and posterior lung involvement was present in 93% of patients. Bilateral pneumonia occurred in 91% of patients. Cellina et al. [42] concluded that GGO was more common bilaterally in the peripheral lung areas under the pleura on CT images of COVID-19. During the disease, the number of consolidations increases, forming fibrotic stripes. Consolidation refers to accumulating fluids, fibrin, and cellular components in the alveolar airspaces. It reduces alveolar air content and increases parenchymal density. Wang et al. [43] found that CT manifestations of mild/common-type infection were multifocal lesions, GGO, involving multiple segments or lobes. CT manifestations of heavy/critical-type infection show consolidation of multiple lesions and interlobular septal thickening. In the paper [44], Bernheim et al. mentioned that the CT image features of COVID-19 are consolidative pulmonary opacities and bilateral and peripheral ground glass. The longer the onset of symptoms, the more findings on CT images. These findings include bilateral and peripheral disease, consolidation, linear opacities, greater total lung involvement, the “reverse halo” sign, and a “crazy-paving” pattern. In Guan’s study [2], 56.4% of the COVID-19 subjects showed GGO.

### 2.2. Chest X-ray

X-rays are emitted from one end, passed through the body, and picked up by a detector at the other end. What is received is a two-dimensional image. So X-ray imaging is fast and cheap. Chest X-ray images of patients diagnosed with COVID-19 show air space consolidation and the bilateral distribution of peripheral hazy lung opacities [45]. When COVID-19 is suspected, the preferred imaging modality for the chest is the X-ray [46]. The radiation dose of chest X-rays is lower than CT, and chest X-rays are cheaper [47]. Because portable chest X-ray is easy to carry and clean, chest X-ray can reduce the risk of COVID-19 transmission during testing [48]. According to Oh [49], although the sensitivity of chest X-ray results is not high (69%) [45], chest X-rays can be used to determine the sequence of treatment for patients infected with COVID-19. Diagnosing chest X-rays can alleviate the saturated healthcare system during the COVID-19 pandemic.

## 3. Convolutional Neural Networks

Convolutional neural networks (CNN) are widely applied to computer vision currently. In the aspect of medical image processing about health, CNN performs outstanding [50]. CNN uses multi-layer superposition to extract from low-level features to high-level features. It is like the hierarchical structure of the human brain function [51,52,53,54]. A CNN comprises an input layer, an output layer, convolution layers, pooling layers, and fully connected layers. Input the raw data in the input layer. Convolution operations are performed to extract features in the convolution layers. The scale of parameters is further reduced in the pooling layers. Fully connected layers connect all the features and output them to the classifier.

CNN is a kind of local perception. Divide the entire image into multiple small windows that can overlap locally. The local features of the image are identified by utilizing sliding windows. A window can be regarded as a filter, which is a neuron. The convolutional layer has a set of such filters.

Figure 1 is a 6×6 image. The size of filter one is 3×3. Put filter one in the upper left corner of the 6×6 image to do the convolution operation, and obtain the result −1. The step size of the sliding window is one. Filter one moves one step to the right and then performs the convolution operation, and the result is −2. Do convolution operation step by step from top to bottom and from left to right, and a 4×4 matrix can be obtained as shown in Equation (1). Doing the same operation with filter2 will obtain another different 4×4 matrix.
(1)$$n_{out}=\mathrm{floor}\left(\frac{n_{in}+2*p-k}{s}\right)+1,$$
where $n_{in}$ is the number of input features. p is the size of padding. k is the kernel size. s is the stride. $n_{out}$ is the number of output features.

The dimensionality of the convolution results can be further reduced in the pooling layers. The location of the pooling layer is usually in the middle of the two convolutional layers [55]. Pooling layers reduce the feature model size of the previous convolutional layer, speed up the calculation, and reduce the probability of overfitting [56]. Pooling operations divide the feature image into regions and choose a value in one pooling area to replace. In this way, the data and parameters of the feature image are compressed. Average pooling [57] and max pooling [58] are the two most typical pooling operations.

The fully connected layers come after the convolution layers and the pooling layers. There may be one or more fully connected layers [59,60,61]. In the fully connected layers, each neuron is fully connected to all neurons in the previous layer. The local features of the previous outputs are integrated into global features through fully connected layers.

### 3.1. Batch Normalization

A deep neural network may have many hidden layers. When training each layer, the parameters are updated, causing the input data distribution of the upper layer to change accordingly. Layer-by-layer accumulation will cause the input distribution of the front layer to change drastically. There will be changes not only in the input layer but also in the hidden layers. The phenomenon is defined as Internal Covariate Shift (ICS) [62]. The front layers should constantly adapt to the update of the parameters in the last layers, which will reduce the training efficiency of the whole network.

Whitening the neural network’s input layer can make the network training converge faster [63,64]. The whitening operation refers to linearly transforming the input data to reach a normal distribution with means of 0 and variances of 1. Each neuron in hidden layers can be seen as input to the next layer. Scholars can perform a whitening operation on each hidden layer neuron to solve the ICS. Transform increasingly skewed data distributions to more standard normal distributions. This is Batch Normalization (BN). BN is located after the input linear activation and before the nonlinear transformation. That is, BN is performed on the activation values $a_{i}$ of neurons in each hidden layer. If there are n activation values in a mini-batch, the process of BN is as follows [62]:(1)Calculate the mean μ, as shown in Equation (2);
(2)$$\mu =\frac{1}{n}\overset{n}{\underset{i=1}{\sum }}a_{i}$$(2)Calculate the variance $\delta^{2}$, as shown in Equation (3);
(3)$$\delta^{2}=\frac{1}{n}\overset{n}{\underset{i=1}{\sum }}{\left(a_{i}-\mu \right)}^{2}$$(3)Normalize $a_{i}^{*}$, as shown in Equation (4);
(4)$$a_{i}^{*}=\frac{a_{i}-\mu }{\sqrt{\delta^{2}+e}}$$
where *e* is a constant used to ensure numerical stability.(4)Scale and shift, as shown in Equation (5).
(5)$$BN\left(a_{i}\right)=\varphi a_{i}^{*}+\omega ,$$

φ is the parameter that scales the normalized value. ω is the parameter that shifts the normalized value. These parameters are learnable. Simply normalizing the input may lead to changing what the input represents. The addition of these parameters can restore the representation of the network.

### 3.2. Dropout Technology

Between the input and the output layers, there are hidden layers. A deep neural network (DNN) usually has many hidden layers. The training set can be modeled correctly by adapting the weights on the incoming connections of the neurons in hidden layers [60]. However, the weight matrix performs poorly on the test set and has poor generalization ability. This phenomenon is called overfitting.

Dropout can solve the problem of overfitting. Dropout means that some neurons in the network are dropped out. However, instead of being deleted, they are temporarily dropped out of the network, including their output and input connections [65], as shown in Figure 2. The neurons that are dropped out are randomly selected. Each neuron is left with a fixed probability p. The neurons that are left form new networks thinner than the original network. These networks use the same weight matrix, so there is no increase in the number of parameters.

When the network is training, a neuron is left with probability p (p is randomly generated from a Bernoulli distribution). It is connected to the next layer with a weight w. Then the weight value is ultimately p×w. To make the weights connecting the neurons in the next layer consistent with those during training, the neurons during testing are always present, and the weights are reduced to p×w, as shown in Figure 3.

The dropout operation of the neural network is as Equation (6) [65]:(6)$${\tilde{d}}^{l}=r^{l}\times d^{l}\;c_{i}^{l+1}=w_{i}^{l+1}\times {\tilde{d}}^{l}+b_{i}^{l+1}\;d_{i}^{l+1}=f\left(c_{i}^{l+1}\right)$$
where $r^{l}$ is a vector produced by the probability $p^{l}$ of the neurons in the hidden layer l. $d^{l}$ is the output vector of hidden layer l. $c^{l}$ is the input vector of the hidden layer l. $w^{l}$ and $b^{l}$ are the weights and biases of hidden layer *l*. $f\left(\right)$ is the activation function. ${\tilde{d}}^{l}$ is the output vector of hidden layer *l* with the dropout. ${\tilde{d}}^{l}$ is processed to be used as the input of the hidden layer l+1.

### 3.3. ReLU Function and Its Variants

#### 3.3.1. ReLU

As the activation function, ReLU functions increase the nonlinear relationship between the layers of CNN to complete complex tasks. The neurons of each layer in the convolutional neural network are weighted and shifted, then output to the next layer, as shown in Equation (7).
(7)y=wx+b,
where *x* is the input vector, *y* is the output vector. *w* is the weight vector for the layer, and *b* is the bias vector for the layer. If there are two hidden layers, the input is weighted and biased twice to obtain the output, as shown in Equations (8) and (9).
(8)$$y=w_{2}\left(w_{1}x+b_{1}\right)+b_{2}$$
(9)$$y=w_{2}w_{1}x+w_{2}b_{1}+b_{2}$$

From Equation (9), it can be concluded that multiple hidden layers can be combined into one layer if only linear transformations of weights and biases are applied to the input. The linear expression ability is limited and insufficient to fit nonlinear neural networks. The activation functions process the outputs of the neurons in the previous layers. The processing results are passed as inputs to the neurons of the next layer. Activation functions can add nonlinear factors. Rectified linear unit (ReLU) is a kind of activation function commonly used in CNN. ReLU function is defined as Equation (10):(10)$$\mathrm{ReLU}\left(m\right)=\left\{\begin{array}{c} 0\;\;\;m<0\\ m\;\;\;m\geq 0 \end{array}\right.$$

ReLU is illustrated in Figure 4a. The ReLU function is a piecewise linear function. It has unilateral inhibition, which enables sparse activation of neurons in a deep neural network. Sparsity networks are more helpful for mining features. The ReLU function accelerates the convergence, and the gradient vanishing problem is solved [66]. A large parameter update may cause some ReLU neurons to never be activated. The phenomenon of “dying ReLU” appeared [67]. Variants of the ReLU function appeared to solve the problem, such as LReLU (Figure 4b), PReLU (Figure 4c), RReLU (Figure 4d), ELU (Figure 4e), etc.

#### 3.3.2. Leaky ReLU

Leaky ReLU (LReLU) solves the dying ReLU problem using a small slope [68]. The LReLU function is defined as Equation (11):(11)$$\mathrm{LReLU}\left(m\right)=\left\{\begin{array}{c} \beta m\;\;\;m<0\\ m\;\;\;\;m\geq 0 \end{array}\right..$$
where β is generally 0.01. The LReLU has the advantage that if the input of the LReLU activation function is less than zero, the gradient can also be calculated, unlike ReLU, which keeps the gradient zero.

#### 3.3.3. Parametric ReLU

Parametric rectified linear unit (PReLU) is a variant of LeakyReLU. The parameter β of PReLU [69] is not set artificially but is obtained through training. The parameter is the key to improving the classification performance. PReLU is defined as Equation (12):(12)$$\mathrm{PReLU}\left(m\right)=\max\left(0,m\right)+\beta \min\left(0,m\right)$$

When β=0, PReLU becomes ReLU. β controls the slope of the negative axis.

#### 3.3.4. Randomized Leaky ReLU

Randomized leaky rectified linear unit (RReLU) is another variant of LeakyReLU. RReLU is defined as Equation (13). Parameter β is randomly valued during training and becomes a fixed value during testing. During the training process, the distribution of β satisfies the normal distribution with a standard deviation of 1 and a mean of 0. When testing, Srivastava [70] et al. took an average of all β using the method of dropout. The NDSB competitor suggested that it would be better if β is sampled from *C* (3,8).
(13)$$\mathrm{RReLU}\left(m\right)=\left\{\begin{array}{c} \beta m\;\;\;m<0\\ m\;\;\;\;m\geq 0 \end{array}\right.\;\;\;\beta \sim C\left(x,y\right),\;x<y\;and\;x,y\in \left[0\right.,1)$$

#### 3.3.5. Exponential Linear Unit

The exponential linear unit (ELU) is also an activation function. It has similarities and differences with ReLU. ELU is an activation function with negative values, and it is defined as Equation (14):(14)$$\mathrm{ELU}\left(m\right)=\left\{\begin{array}{c} m\;\;\;\;\;\;\;\;m>0\\ \beta (e^{m}-1)\;m\leq 0 \end{array}\right.,\;\;{\mathrm{ELU}}^{\prime }\left(m\right)=\left\{\begin{array}{c} 1\;\;\;\;\;\;\;\;\;\;m>0\\ ELU\left(m\right)+\beta \;m\leq 0 \end{array}\right.\;\;(\beta >0)$$
where β controls the saturation value for negative net input [71]. ELU solves the problem of gradient vanishing. The mean values of the ELU output are close to zero, which makes the convergence faster. Because ELU is nonzero for negative values, there is no “dying ReLU” phenomenon. ELU is an unsaturated function, and there is no problem with vanishing gradients or exploding gradients.

### 3.4. Pooling

Pooling usually follows the convolution operation. The essence of pooling is sampling. The dimension of the input feature map can be reduced using pooling by choosing an appropriate method. Pooling can reduce the operation’s complexity and improve the operation’s speed. The addition of pooling layers in the architecture of CNN can reduce the scale of weights and parameters [72], reducing the input’s dimensions and memory consumption [73,74,75]. The use of pooling layers can also solve overfitting [76].

#### 3.4.1. Max Pooling

Max pooling is an extensively applied pooling method in CNNs [58]. This method is popular because it does not require tuning parameters. The max pooling technique is to find the max element in the pooling region [77,78]. Max pooling can exhibit the max value of the feature map in the k×k neighborhood. Not only the scale of the feature map space is optimized, but also the translation invariance of the network is preserved by max pooling [79].

In Max pooling, the whole image is divided into several blocks of the same size that do not overlap. We discard all other nodes and take only the max value in each block. For example, there is a pooling region of size 4×4. The pooling filter size is 2×2, and the stride is 2. These setups enable pooling to be applied to regions of the image that do not overlap. The pooling filter selects the max value of each block to obtain the final output. Figure 5 shows the process of max pooling. Max pooling focuses on the max element and discards the others. This may lead to wrong results of the disappearance of salient features [80].

#### 3.4.2. Average Pooling

The input sample is divided into multiple blocks of the same size and not overlapping each other. Average pooling is to calculate the average value of all elements in each block and present output. Use the input sample in Section 3.4.1. Similarly, the average pooling filter size is 2×2, and the stride is 2. The purpose is to ensure that the areas swept by the pooling filter do not overlap. Calculate the average of all values in each 2×2 pooling block as the output. Figure 6 shows the process of average pooling. In average pooling, if the averages are low, the contrast will be diminished [79]. The convolutional features will be decreased [81] if most of the averages are zero. Wang et al. [82] proposed using a rank-based average pooling module in the network for COVID-19 COVID-19 recognition to avoid overfitting.

#### 3.4.3. Fractional Max Pooling

Fractional max pooling (FMP) is a special max pooling. It has an important parameter μ called the pooled fraction. Because μ represents the ratio of the input-spatial size to the output-spatial size of the pooling region, the image size can be reduced by setting μ. For regular max pooling, μ is set to an integer. The size of the feature map is rapidly reduced. For fractional max pooling, μ is set to a non-integer (a fraction). The decay rate of the feature map will slow down. If the focus is on accuracy, set $\mu \in \left(1,2\right)$, as shown in Figure 7. Suppose the focus is on speed, set $\mu \in \left(2,3\right)$ [83].

There are two types of pooling regions: disjoint rectangular areas and overlapping square areas. The pooling regions can be chosen in pseudorandom and random ways. FMP works better in a random way [83]. Wang et al. [84] used fractional max pooling (FMP) instead of max pooling and average pooling in a network for COVID-19 recognition.

#### 3.4.4. Other Popular Pooling Methods

Both average pooling and max pooling have disadvantages. Yu et al. [85] proposed a way to mix the two. This method is called mixed pooling. The weights of average pooling and max pooling are merged [86], which can overcome their shortcomings. In many experiments, the performance of mixed pooling is better than that of max pooling and average pooling in image classification [87,88,89].

Tree pooling [90] uses the data from pooling filters to learn and combines these learned filters. The performance of tree pooling is better than that of average pooling, max pooling, and mixed pooling. However, more parameters must be learned than mixed average and max pooling [91]. Tree pooling is suitable for lower network layers focusing on functional responses [92].

Fergus and Zeiler et al. [59] proposed stochastic pooling. Stochastic pooling randomly selects values. It is not suitable for negative activations. With little training data, stochastic pooling tends to generate strong activation, leading to overfitting. Rank-based stochastic pooling [93] may solve this problem. It estimates the ranking of the activations within the pooling region. Stochastic pooling needs to address issues related to scaling [94]. Wang et al. [95] proposed the stochastic pooling module and a stochastic pooling neural network for COVID-19 diagnosis. Zhang et al. [96] proposed a deep-learning model for COVID-19 diagnosis. It used stochastic pooling instead of max pooling and average pooling.

## 4. Data Augmentation

Data augmentation (DA) is an effective way to expand datasets. The larger the size of the data, the better the generalization ability of the trained model. When the number of samples in the dataset is relatively small, data augmentation increases the amount of training data by modifying the existing training data [97]. Data augmentation is mainly used to prevent overfitting, especially when the training dataset is small. There are many ways for data augmentation [98,99,100,101].

### 4.1. Geometric Transforms

Geometric transforms are standard methods of data augmentation. Training data can be effectively increased with geometric transforms [102]. Flipping is one of the easiest ways to implement data augmentation [103]. Flipping is a mirror image reflected along a line. Horizontal axis flipping is commonly used. Figure 8a shows the original image. Figure 8b shows the image after horizontal flipping. Figure 9a shows the original image. Figure 9b shows the image after vertical flipping. No matter what direction the image is flipped, the image remains the same.

Rotation augmentation rotates the image to the left or right according to the selected angle. The image is rotated around a central point [104]. The angle of rotation should be chosen appropriately. A wide rotation may make recognition unsafe [105]. Figure 10a shows the original image. Figure 10b shows the result of a clockwise rotation of 10 degrees.

Shear is the non-vertical projection effect of a plane object on the projected plane. Horizontal shear and vertical shear are commonly used in data enhancement. If the original coordinate of a point is $\left(x,\;y\right)$, the coordinate after horizontal shear is $\left(x_{h}^{\prime },y_{h}^{\prime }\right)$, as shown in Equation (15),
(15)$$\left[x_{h}^{\prime },\;y_{h}^{\prime },1\right]=\left[x,y,1\right]\times \left[\begin{array}{ccc} 1 & 0 & 0\\ \beta_{h} & 1 & 0\\ 0 & 0 & 1 \end{array}\right]$$
where $\beta_{h}$ is horizontal shear parameter. The coordinate after the vertical shear is $\left(x_{v}^{\prime },y_{v}^{\prime }\right)$, as shown in Equation (16), where $\beta_{v}$ is vertical shear parameter.
(16)$$\left[x_{v}^{\prime },\;y_{v}^{\prime },1\right]=\left[x,y,1\right]\times \left[\begin{array}{ccc} 1 & \beta_{v} & 0\\ 0 & 1 & 0\\ 0 & 0 & 1 \end{array}\right]$$

Cropping cuts a portion of the input image. It is an effective tool for extracting patches [106,107]. Images can be classified according to patches rather than the whole image. An image can be cropped into several pieces. It does not lose important information about the image when cropping.

### 4.2. Noise Injection

There are many ways for data augmentation using noise injection, such as Gaussian noise, salt-and-pepper noise, speckle noise, etc. Noise injection has been successfully used in plant leaf disease recognition [108], robot speech commands [109], fruit classification [110], and so on. Gaussian noise injection is more popular. Gaussian noise follows the normal distribution. The probability density function (PDF) $f\left(x\right)$ of Gaussian noise can be expressed as Equation (17), where $\alpha^{2}$ is the variance and β is the mean value. Gaussian injection means injecting random values from a Gaussian distribution into the pixels of the image.
(17)$$f\left(x\right)=\frac{1}{\sqrt{2\pi \alpha^{2}}}e^{-\frac{{\left(x-\beta \right)}^{2}}{2\alpha^{2}}}$$

In addition to adding noise to the input layer, you can also add noise to other layers. DeVries et al. add noise to a learned feature space instead of the input space [111]. Xie et al. [112] add noise to the loss layer. These prevent the network from overfitting.

### 4.3. Color Space

Implementing data augmentation in a color space is also a practical approach. The transformations are based on gray or RGB color values [113]. Color augmentation can be completed by isolating a color channel and converting an image into a representation in one color channel. Color augmentation can be completed by manipulating the RGB values to change the image’s brightness. Color augmentation can be completed by changing the color histograms of the image [98].

### 4.4. Random Erasing

Random erasing means randomly selecting an area of the image, and its pixels are erased by random or mean pixel values [114]. It can augment the recognition of occluded images. Random erasing requires no parameters to learn and is easy to implement.

The size of the source image is M×N. The size of the randomly selected erasing rectangle region is $M^{\prime }\times N^{\prime }$. Then, select randomly a point $W=\left(x,y\right)$ in the source image region. If $x+M^{\prime }<M$ and $y+N^{\prime }<N$, $S=\left(x,y,x+M^{\prime },y+N^{\prime }\right)$ is the erasing region. Each pixel in *S* is assigned a random value between 0 and 255. Finally, a new image with a part of the regions erased is obtained.

### 4.5. Kernel Filters

Kernel filters can sharpen or blur images. The data augmentation can be obtained by convolving the kernel filter with the image. A Gaussian blur filter [115] is used to slide an n×n matrix across the image. A blurry image is yielded. An unsharp masking [116] slides an n×n matrix over the image. A sharpened image is yielded. The blurred image can resist motion blur. The sharpened image allows for more detail. Figure 11a shows the original image. Figure 11b shows the image after sharpening. Figure 11c shows the image after blurring.

### 4.6. Mixing Images

Mixing images is a method for data augmentation of two or more images. Calculating the average of the pixels can achieve a mixed image. This strategy of data augmentation may seem irrational, but it is effective. Ionue [117] came up with Samplepaining technology. Two images were randomly selected from the dataset, and one was overlaid on another. A new sample is obtained by superimposing two images. This simple data augmentation technology significantly improved classification accuracy. Summers and Dinneen [118] mixed the images in a nonlinear manner. Takahashi et al. [119] generate a new training image by mixing four images obtained through random image cropping and patching (RICAP).

### 4.7. Data Augmentation Methods Based on Deep Learning

There are some DA ways based on deep learning. The feature space in CNN is a low-dimensional representation in high-level layers [98]. Feature space augmentation opens up opportunities for many vector operations for data augmentation [111]. Adversarial training helps to search the space for possible augmentations. It can improve the weakness in the boundaries of learnable decisions. Generative Adversarial Networks (GANs) are a way to obtain additional information from a dataset [120]. Figure 12 shows the architecture of GAN.

Neural style transfer transfers the style of an image created in CNN to another image and preserves the original content [121]. Overfitting of the model can be prevented. Meta-learning refers to optimizing neural networks with neural networks [122]. There is neural augmentation [123], smart augmentation [124], and auto-augmentation [125].

## 5. Pretrained Models

With the development of CNN, many mature networks perform well in image recognition, object recognition, natural language processing, and other fields. These networks include AlexNet, ResNet, DenseNet, VGG, GoogleNet, etc. During the COVID-19 pandemic, these networks could be fine-tuned and used to classify chest CT or chest X-ray images into different categories (such as infection and health).

### 5.1. AlexNet

AlexNet contains five convolutional layers and three fully connected layers [66]. Compared with LeNet, it uses more new technology [126]. Figure 13 is the structure of AlexNet.

(1)AlexNet uses ReLU as the activation function. ReLU solves the problem of gradient descent.(2)For fully connected layer training, AlexNet randomly ignores some neurons using dropout to avoid model overfitting.(3)AlexNet uses overlapping max pooling in CNN. The step size in AlexNet is smaller than the size of the pooling kernel, so overlapping pooling is obtained.(4)AlexNet uses the local normalization scheme, which is more helpful for generalization.(5)AlexNet uses the computing power of parallel GPU, and the training of CNN is accelerated.(6)Two different forms of data augmentation are used in AlexNet. The first form of data augmentation is the translation and horizontal reflection. The second form of data augmentation is the change in the intensity of the image color channel.

### 5.2. ResNet

ResNet is developed and optimized based on AlexNet. One of the advantages of residual neural networks is identity mapping. ResNet has 152 layers. Simply increasing the depth of the network can lead to the problem of vanishing gradients or exploding gradients [127,128]. This problem has been addressed by normalized initialization [63,69,128,129] and intermediate normalization layers. However, with the network layers stacking, the accuracy of the training dataset saturates or even descends. That is the degradation. The degradation is not due to overfitting. Stacking layers in the deep model will make the training error larger [130,131]. ResNet uses a deep residual learning framework to mitigate the degradation [132].

The shallow network gradually stacks layers to make it a deep network. In a deep network, if the stacked layers are identity mapping, and the other layers copy the shallow network, the performance can be almost the same as the shallow network. A few stacked layers are called a block. *x* is the input to the first layer of the block. The expected mapping is $G\left(x\right)$. The fitting function of the block is Equation (18):(18)$$D\left(x\right)=G\left(x\right)-x.$$

So $G\left(x\right)$ is recast into $D\left(x\right)+x$. It is easier to approximate $D\left(x\right)$ to $G\left(x\right)-x$ than to approximate $D\left(x\right)$ to $G\left(x\right)$. $G\left(x\right)-x$ is the residual mapping. To the extreme, the identity mapping$\;G\left(x\right)=x$ can be obtained by pushing the residual $D\left(x\right)$ to zero. Figure 14 shows that feedforward neural networks with shortcut connections can achieve $D\left(x\right)+x$.

### 5.3. DenseNet

All the front layers in DenseNet are densely connected to the back layers. Each layer connects to every other layer. One of its characteristics is that the connection of features on channels enables feature reuse. DenseNet has fewer parameters, faster computation, and better performance.

DenseNet is a densely connected convolutional network. Feature maps from all preceding layers are concatenated as additional inputs to this layer. Its feature maps are passed on as inputs to all subsequent layers, which preserves the feedforward nature of the network. If DenseNet has *N* layers, there are $N\left(N+1\right)/2$ connections. $x_{0}$ is the input image of a DenseNet. $x_{n}$ is the output of the network at layer *n*. Layer *n* receives feature maps from all preceding layers as inputs (19):(19)$$x_{n}=F_{n}\left(\left[x_{0},\;x_{1},\cdots ,x_{n-1}\right]\right),$$
where $F_{n}()$ is a nonlinear transformation function of layer n. It is a composite function of operations [133]. It includes batch normalization (BN), rectified linear units (ReLU), pooling [126], or convolution.

Pooling and convolution in CNN will change the size of feature maps. In DenseNet, only feature maps of the same size can be densely connected. To resolve this contradiction, DenseNet divides the network into several dense blocks. The feature maps of each layer in one dense block have the same size. They are densely connected. Between two adjacent dense blocks are transition layers. Pooling and convolution in the transition layer make the feature map smaller. Figure 15 is the structure of DenseNet.

### 5.4. VGG

Visual Geometry Group (VGG) won first place in localization and second place in the classification on ImageNet Challenge in 2014. VGG is an improvement on AlexNet by deepening the depth of the network. VGG uses 3×3 convolution filters in all layers. It enables the network to add more convolutional layers. Increasing the depth of the network leads to better performance. Figure 16 is the structure of VGG16.

VGG uses a stack of three 3×3 convolution kernels. Three 3×3 convolution kernels have the same receptive field of 7×7. The three nonlinear rectification layers make the decision function more discriminative. Then, the number of parameters is decreased. If the input and the output of three 3×3 convolution layers both have *X* channels, the number of parameters is $3\times 3^{2}\times X^{2}=27X^{2}$. One 7×7 convolution layer has the parameters is $7^{2}\times X^{2}=49X^{2}$, 81% more parameters [61].

VGG uses the max pooling layer. The size of the pooling kernel is 2×2, with stride 2. Smaller pooling kernels can capture more details of the information. Max pooling is easier to capture the changes in the image and obtain more differences in local information.

### 5.5. GoogleNet

GoogleNet believes that increasing the number of layers (depth) or the number of neurons at each layer (width) of the networks can improve the performance of the networks. Increasing the size of a network has two drawbacks:(1)Larger sizes of networks generate more parameters. When the data in the training set is small, too many parameters will cause the network to overfit.(2)Larger size of networks increases computation dramatically. Adding a convolution layer will result in a quadratic increase in computation.

To overcome the two drawbacks, GoogleNet uses a sparse connection instead of a full connection. Inception architecture is constructed in GoogleNet to realize sparse connections [134]. The idea of Inception is to find the optimal local sparse structure in the network and repeat it. The filter sizes of Inception are 1×1, 3×3, and 5×5. The output filter banks concatenate into a single vector as the input of the next step. At the same time, it is also necessary to add an alternative parallel pooling path in each stage, as shown in Figure 17.

But in Inception architecture, the outputs of the convolutional layer and the pooling layer will increase. It can become expensive and a computational blow-up. Dimension reductions and projections can solve this problem. Using 1×1 convolutions can reduce the computation before 3×3 and 5×5 convolutions and rectify linear activation, as shown in Figure 18.

### 5.6. Transfer Learning

In some cases, it is not easy to obtain training data that matches the feature space of the test data [135]. If a test dataset is related to a training dataset that has already been trained, the parameters of the trained model (pre-trained model) can be transferred to the target model, as shown in Figure 19. This avoids training the target model from scratch. It speeds up and optimizes the learning of the target model. Transfer learning can provide related but not identical existing datasets for the target model.

Transfer learning has been applied to include text sentiment classification [136], image classification [137,138,139], human activity classification [140], software defect classification [141], and multi-language text classification [142,143,144]. The networks that can be used as pre-trained are also the focus of research.

The feature space of the source domain is $F_{s}$, and the feature space of the target domain is $F_{t}$. If $F_{t}=F_{s}$, this is homogeneous transfer learning. If $F_{t}\neq F_{s}$, this is heterogeneous transfer learning. The strategies adopted by homogeneous transfer learning [135] include correcting the difference in the marginal distribution in the source, correcting the difference in the conditional distribution in the source, or correcting the difference between the marginal and conditional distribution in the source. Heterogeneous transfer learning [135] uses strategies that align the input spaces of the source and target domains. If the domain distributions are still unequal, further modulations are needed.

There are three strategies for transfer learning: (i) inductive transfer learning, (ii) transductive transfer learning, and (iii) unsupervised transfer learning [145,146]. There are two common types of inductive learning. One is to use the source domain to obtain a trained learning model and fine-tune different target layers [147]. Another is multi-task—learning multiple tasks simultaneously from the same input [148]. The target task is similar to the source task in transductive transfer learning, but the domains differ. The factors affecting transductive transfer learning include domain adaptation [149], sample selection bias [150], and covariate shift [151]. Unsupervised transfer learning mainly solves unsupervised learning tasks in the target domain.

There are four kinds of approaches to transfer learning [145]. The first is instance-based transfer learning. A common approach is to reweight instances of the source domain [152,153,154]. These worked best when the conditional distributions were the same in the source domain and target domain. The second is feature-based transfer learning. The standard methods include asymmetric feature transformation [155] and symmetric feature transformation [156]. The third is parameter-based transfer learning. Source tasks and target tasks share the parameters of the models [157,158]. Target learners are formed by optimally combining the reweighted source learners [158,159,160]. The last one is relational-based transfer learning [161,162,163]. If the source and target data have similar relationships, the relationship among the data can be transferred. Figure 20 shows the classification of transfer learning.

## 6. Application, Analysis, and Discussion

Currently, the diagnosis of COVID-19 mainly relies on epidemiological history, clinical symptoms, laboratory results, chest imaging findings, nucleic acid testing, or homologous comparison of gene sequencing [164]. COVID-19 is highly contagious and has an incubation period. It needs repeated testing over some time to determine infection. It is difficult for doctors to diagnose COVID-19 quickly and accurately. Chest computed tomography (CT) and chest X-ray images can reflect whether the patient’s lung is infected and the degree of damage. However, doctors rely on experience to judge CT images, and X-ray images can also lead to misdiagnosis. Popular artificial intelligence (AI) systems can quickly analyze large amounts of image data. After learning, the system can accurately classify chest CT images and X-ray images. This will help doctors diagnose COVID-19 quickly and accurately.

### 6.1. CNN Applied to CT Images

The papers mentioned in Table 2 used convolutional neural networks (CNN) to analyze chest CT images and proposed various algorithms for COVID-19 diagnosis. These algorithms modify the parameters and architecture of the convolutional neural network (CNN) to obtain better accuracy.

Aslan et al. [165] improved the architecture of AlexNet to obtain mAlexNet. In the mAlexNet architecture, the fully connected layer eight has 25 neurons instead of 1000 neurons. The features extracted by mAlexNet are given into the tree seed algorithm (TSA) ANN structure for classification. The experimental comparison shows that the performance of mAlexNet + TSA-ANN is more excellent. AlexNet and mAlexNet can also obtain good performance by hybridizing other structures or algorithms [166,167]. Different total number of layers [168,169] in ResNet leads to different performance results. Rahimzadeh et al. [170] modified ResNet50V2. This model can retain the data of small objects and improve the classification accuracy of images containing small important objects. By comparison, Loey et al. [171] found that among AlexNet, VGGNet16, VGGNet19, GoogleNet, and ResNet50, the ResNet50 was the most appropriate deep learning model for using the classical data augmentation and CGAN. Özdemir et al. [97] extended ResNet50 architecture with a feature-wise attention layer and used the mixup data augmentation technique. This architecture achieves higher accuracy. Mondal et al. [172] proposed the scheme of optimized InceptionResNetV2 for COVID-19 (CO-IRv2). It combines InceptionNet with ResNet with hyperparameter tuning. DenseNet-121 is a network with 121 layers [173]. DenseNet-121 D solves the problem of vanishing gradients, allows better feature reuse, and reduces the number of parameters. It is more beneficial to the training of deep learning models [174]. Xiao et al. [175] improved DenseNet with a parallel attention module (PAM-DenseNet) which has spatial and channel attention modules. These make the net have better classification performance and patient-wise prediction performance. The VGG network alone [176], or together with other neural networks [177], has been used to classify chest CT images with high accuracy. Wang et al. [178] proposed a VGG-style base network (VSBN). Convolutional block attention module (CBAM) serves as the attention module of VSBN. In order to solve the problem of artificial intelligence (AI) model overfitting, VSBN uses an improved multiple-way data augmentation method. GoogleNet is retrained over COVID-CT-Dataset [179]. GoogleNet learned the variations present in diverse types of CT images. The system model GoogleNet-COD developed by Yu et al. [180] takes GoogleNet as the backbone network. It removes the last top-two layers and replaces them with four new layers, which include the dropout layer, two fully connected layers, and the output layer. Zhang et al. improved the deep convolutional neural network (DCNN) [181]: the pooling layer adopts stochastic pooling; construct a convolution block (CB), which is obtained by combining the convolution layer with the batch normalization layer; construct a fully connected block (FCB), which is obtained by combining the dropout layer with the fully connected layer. Pham [182] uses multiple CNNS to classify CT images collected from COVID-19 patients and non-COVID-19 subjects. Among them, the deepest net, DenseNet201, has the best performance. Transfer learning with the direct input of whole image slice and without using data augmentation provided better classification rates. JavadiMoghaddam et al. [183] proposed a convolutional neural network structure containing a wavelet and four convolutional layers. It optimizes convergence time using batch normalization (BN) and Mish Functions. The Haar wavelet transform occurs at the pooling layer. There are squeeze and Excitation blocks (SE blocks) after each dropout layer.
jimaging-09-00001-t002_Table 2Table 2Performances of CNN in COVID-19 Diagnosis using CT images.ReferenceMethodPerformancesDatasets[165]mAlexNetAccuracy: 97.92%, Sensitivity: 98.20%, Specificity: 97.68%, Precision: 97.32%, F1 score: 97.76%SARS-CoV-2 Ct-Scan Dataset: 2482 chest CT scans[165]mAlexNet + TSA-ANNAccuracy: 98.54%, Sensitivity: 97.75%, Specificity: 99.23%, Precision: 99.09%, F1 score: 98.41%[166]mAlexNet + BiLSTMAccuracy: 98.70%COVID-19 Radiography Database[167]DC-Net-RVFLAccuracy: 90.91%, Sensitivity: 85.68%, Specificity: 96.13%, Precision: 95.70%, F1 score: 90.41%A Private Dataset: 296 lung window images[168]ResNet18Accuracy: 86.70%, Precision: 80.80%, Recall: 81.50%, F1 score: 81.10%A Private Dataset: 618 chest CT samples[169]ResNet50Accuracy: 76%, Specificity: 61.50%, Recall: 81.10%, AUC: 0.8190A Private Dataset: 495 chest CT images [170]Modified ResNet50V2Accuracy: 98.49%, Recall: 96.83%COVID-Ctset: 63849 chest CT images[171]ResNet50 + Data augmentations + CGANAccuracy: 82.91%, Sensitivity: 77.66%, Specificity: 87.62%.COVID-19 CT Scan Digital Images Dataset: 742 chest CT images[97]ResNet50+Attention+mixupAccuracy: 95.57%COVID-CT Dataset: 1596 chest CT images[172]CO-IRv2 AdamAccuracy: 94.97%, Specificity: 96.52%, Precision: 96.90%, F1-Score: 95.24%, Recall: 93.63%, Execution Time(sec): 717A New Dataset: 2481 chest CT images[172]CO-IRv2 NadamAccuracy: 96.18%, Specificity: 95.08%, Precision: 95.35%, F1-Score: 96.28%, Recall: 97.23%, Execution Time(sec): 707[172]CO-IRv2 RMSPropAccuracy: 96.18%, Specificity: 99.18%, Precision: 99.16%, F1-Score: 96.13%, Recall: 93.28%, Execution Time(sec): 749[173]DenseNet-121Accuracy: 92%, Recall: 95% A Real Patient Image Dataset: 2482 chest CT images[175]PAM-DenseNetAccuracy: 94.29%, Sensitivity: 95.74%, Specificity: 96.77%, Precision: 93.75%Dataset 1: A Lung CT Slices Dataset, 3530 chest CT slicesDataset 2: A Lung CT Scans Dataset, 280 chest CT scans[176]VGG-19Accuracy: 94.52%COVID-19 CT Dataset: 738 chest CT scan images[177]SRGAN +VGG16Accuracy: 98.0%, Sensitivity: 99.0%, Specificity: 94.9%COVID-CT-Dataset: 470 chest CT images[178]AVNCThe sensitivity, precision, F1 all above 95%A Private Dataset: 1164 slice images [179]GoogleNetAccuracy: 82.14%COVID-CT-Dataset: 349 chest CT images[180]GoogleNet-CODAccuracy: 87.50%, Sensitivity: 90.91%, Specificity: 84.09%A Private COVID-19 Dataset: 148 chests CT images [181]5L-DCNN-SP-CAccuracy: 93.64%, Sensitivity: 93.28%, Specificity: 94.00%, Precision: 93.96%, F1 score:93.62%A Private Dataset: 320 chest CT images[182]AlexNetAccuracy: 86.85%, Sensitivity: 80.25%, Specificity: 94.29%, F1 score: 0.85, AUC: 0.94COVID-CT-Dataset: 349 chest CT images[182]GoogleNetAccuracy: 93.83%, Sensitivity: 96.71%, Specificity: 90.57%, F1 score: 0.94, AUC: 0.96[182]ResNet-18Accuracy: 95.44%, Sensitivity: 98.99%, Specificity: 91.43%, F1 score: 0.96, AUC: 0.98[182]ResNet-50Accuracy: 93.62%, Sensitivity: 95.57%, Specificity: 91.43%, F1 score: 0.94, AUC: 0.98[182]ResNet-101Accuracy: 93.29%, Sensitivity: 96.20%, Specificity: 90.00%, F1 score: 0.94, AUC: 0.98[182]Inception-ResNet-v2Accuracy: 88.59%, Sensitivity: 89.24%, Specificity: 87.86%, F1 score: 0.89, AUC: 0.96[182]VGG-16Accuracy: 89.26%, Sensitivity: 92.83%, Specificity: 85.24%, F1 score: 0.90, AUC: 0.96[182]VGG-19Accuracy: 90.16%, Sensitivity: 87.34%, Specificity: 93.33%, F1 score: 0.90, AUC: 0.97[182]DenseNet-201Accuracy: 96.20%, Sensitivity: 95.78%, Specificity: 96.67%, F1 score: 0.96, AUC: 0.98[183]WCNN4Accuracy: 99.03%COVID-19 CT Dataset: 19685 chest CT images


As can be seen from Table 2, there have been many suggestions, proposals, and implementations for applying CNN to COVID-19 diagnosis by analyzing chest CT images. For example, some researchers focus on adjusting the architecture of the network. Some focus on adjusting the number of layers of the network, some on improving existing algorithms, and some on mixing several structures.

Many schemes listed above have achieved accuracy higher than 90% when implemented. However, there are a lot of CNN networks that have a huge space for improvement:(1)It can be concluded from the results that the modified AlexNet can obtain better accuracy. When combined with other structures, the network is improved so that the accuracy is higher for COVID-19 diagnosis.(2)Different accuracy will be obtained from different depths of ResNet. Common ones include ResNet18, ResNet50, ResNet101, etc. However, according to the results of the above experiments, ResNet18 performs the best. ResNet can be modified to improve accuracy. The modified ResNet or ResNet combined with other networks could achieve more than 90% accuracy. ResNet50 is often used with other architectures or algorithms for COVID-19 diagnosis. Moreover, we found that the larger the dataset ResNet applied, the higher the accuracy.(3)With DenseNet, the accuracy was close to 95%. DenseNet has a better performance than other networks on the same dataset. The dataset size does not affect DenseNet accuracy as much as the depth of the network. DenseNet201 shows excellent performance for COVID-19 diagnosis.(4)Even on small datasets, VGG combined with other structures or algorithms can achieve more than 95% accuracy. VGG19 and VGG16 performed similarly.(5)The accuracy of GoogleNet for COVID-19 diagnosis is not good enough, generally less than 90%. Try tweaking the architecture of GoogleNet or combining GoogleNet with other networks to improve classification accuracy.(6)For almost all networks, accuracy increases as the dataset become larger. Private datasets are generally smaller than public datasets. When there are more than 1000 images in the dataset, almost all networks or models can achieve more than 95% accuracy.

### 6.2. CNN Applied to X-ray Images

The papers mentioned in Table 3 use the architecture or technology of convolutional neural networks (CNNs) to analyze chest X-ray images and establish various training models for COVID-19 diagnosis. Through optimizations, these automatic diagnosis systems have achieved better performance. Chest X-ray images were more readily available. So COVID-19 diagnosis systems could be more widely used by analyzing chest X-ray images in areas with inadequate medical systems.

Cortés et al. [184] applied to learn transfer to AlexNet and fine-tuned it. The first layer of AlexNet is replaced for images in a single intensity. Kaur et al. [185] used the improved AlexNet architecture. Strength Pareto evolutionary algorithm-II (SPEA-II) is used to optimize parameters. Narin et al. [186] compared different layers of ResNet to classify chest X-ray images. ResNet50 provides the highest classification performance. In the study of Chowdhury et al. [187], DenseNet201 performed well in classifying chest X-ray images with image augmentation. In the study, Hernandez et al. applied transfer learning through ResNet, DenseNet, and VGG and fine-tuned them, which achieved higher accuracy [188]. Sitaula et al. [189] added the attention module to the appropriate convolution layer of VGG16. Classification experiments were performed on three COVID-19 chest X-ray image datasets. Haritha et al. [190] used GoogleNet to classify X-ray images and predict COVID-19. Kaya et al. [191] first applied the angle transformation (AT) on X-ray images. Then these images are trained using GoogleNet combined with LSTM. Finally, they obtained a better accuracy rate.
jimaging-09-00001-t003_Table 3Table 3Performances of CNN in COVID-19 Diagnosis using X-ray images.ReferenceMethodPerformancesDatasets[184]AlexNetAccuracy: 96.5%, Sensitivity: 98.0%, Specificity: 91.7%Six Public Databases: 11,312 chest X-ray images[185]mAlexNet + SPEA-IIAccuracy: 99.130%, Sensitivity: 99.476%, Specificity: 99.154%Dataset 1: Kaggle Dataset, 3050 chest X-ray imagesDataset 2: 1203 chest X-ray scans[186]ResNet50Accuracy: 99.7%, Specificity: 99.8%, Precision: 98.3%, F1 score: 98.5%, Recall: 98.8%Dataset 1: GitHub Dataset, 341 chest X-ray imagesDataset 2: ChestX-ray8 Database, 2800 chest X-ray imagesDataset 3: Kaggle Dataset, 4265 chest X-ray images[186]ResNet101Accuracy: 94.7%, Specificity: 99.9%, Precision: 98.9%, F1 score: 68.6%, Recall: 52.5%[186]ResNet152Accuracy: 92.8%, Specificity: 98.0%, Precision: 75.7%, F1 score: 60.9%, Recall: 51.0%[187]DenseNet201Accuracy: 97.94%, Sensitivity: 97.94%, Specificity: 98.80%, Precision: 97.95%, F1 score: 97.94%Dataset 1: COVID-19 Database, 423 chest X-ray imagesDataset 2: 1579 normal chest X-ray images Dataset 3: 1485 viral pneumonia chest X-ray images[188]ResNet50 + fine tuningAccuracy: 90.63%, Precision: 90.00%, F1 score: 90.72%, Recall: 91.67%Italian Society of Medical and Interventional Radiology and ChexPert Dataset, 27000 chest X-ray images[188]DenseNet121 + fine tuningAccuracy: 83.4%, Precision: 89%, F1 score: 76.19%, Recall: 67%[188]VGG16 + fine tuningAccuracy: 82.29%, Precision: 80.39%, F1 score: 82.82%, Recall: 85.41%[189]VGG-16 + attention module + convolution moduleAccuracy: 79.58% (Dataset 1), 85.43% (Dataset 2), 87.49% (Dataset 3)Dataset 1: Public Databases, 1125 chest X-ray imagesDataset 2: Public Databases, 1638 chest X-ray images Dataset 3: Public Databases, 2138 chest X-ray images[190]GoogleNetTraining Accuracy: 99%, Testing Accuracy: 98.5%Public Dataset: 1824 chest X-ray images[191]AT + GoogleNet + LSTMAccuracy: 98.97%Mendeley Database: 1824 chest X-ray images


As can be seen from Table 3, there have also been many suggestions, proposals, and implementations for applying CNN to COVID-19 diagnosis by analyzing chest X-ray images. The researchers focused on tweaking the number of layers in the network and mixing CNNS with other network structures. They divided datasets into several classes for training and validation. Many automatic diagnosis systems can achieve accuracy higher than 90%. However, there is a huge space for improvement.

## 7. Conclusions

In this paper, we reviewed and summarized convolutional neural networks in COVID-19 diagnosis. We introduced various technologies related to CNN and some mature CNN networks with excellent performance. Then, we analyzed and compared various suggestions of other researchers on the application of CNN for COVID-19 diagnosis. Here are a few conclusions and suggestions:(1)At present, rapid and accurate COVID-19 diagnosis is vital. The classification method of chest CT or chest X-ray images based on CNN plays an important role.(2)The current experiment has limited datasets. It is necessary to collect more data or explore better methods for small datasets.(3)Most experiments do not consider the execution time problem. It is necessary to shorten the execution time with appropriate data preprocessing [192,193,194,195] strategies or GPU acceleration.(4)The experiments discussed in this paper use chest CT or chest X-ray images as the input datasets of CNN and have achieved good performance. Although X-ray image is not as good as CT in performance, it has low cost, low radiation dose, and easy-to-operate in general hospitals [196]. Future research could consider more types of medical image forms. The application of the CNN method on medical images has potential value for COVID-19 diagnosis.

## Figures and Tables

**Figure 1 jimaging-09-00001-f001:**
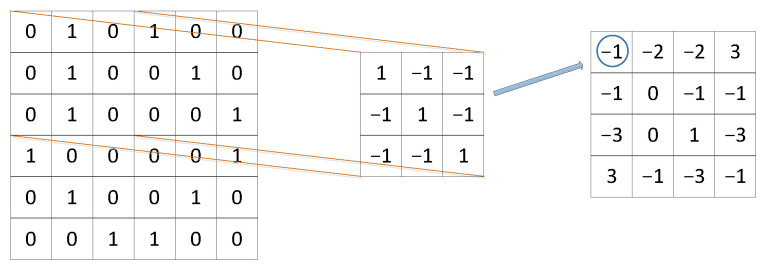
Convolution operation.

**Figure 2 jimaging-09-00001-f002:**
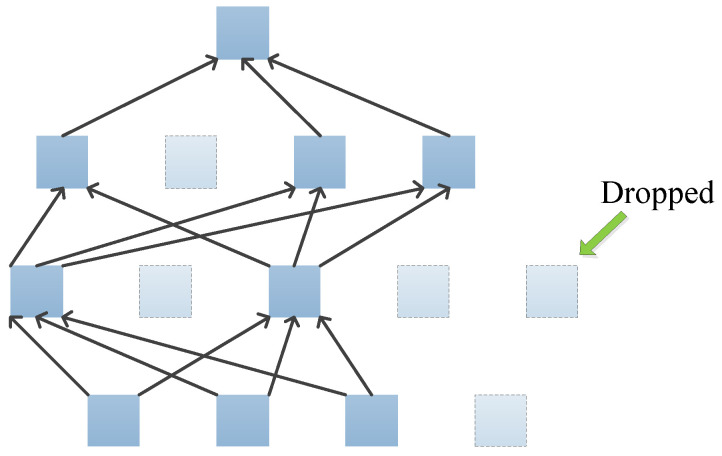
Dropout neural net.

**Figure 3 jimaging-09-00001-f003:**
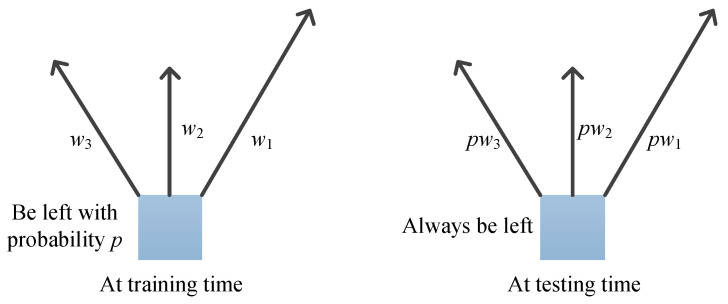
The output during testing is consistent with the output during training.

**Figure 4 jimaging-09-00001-f004:**
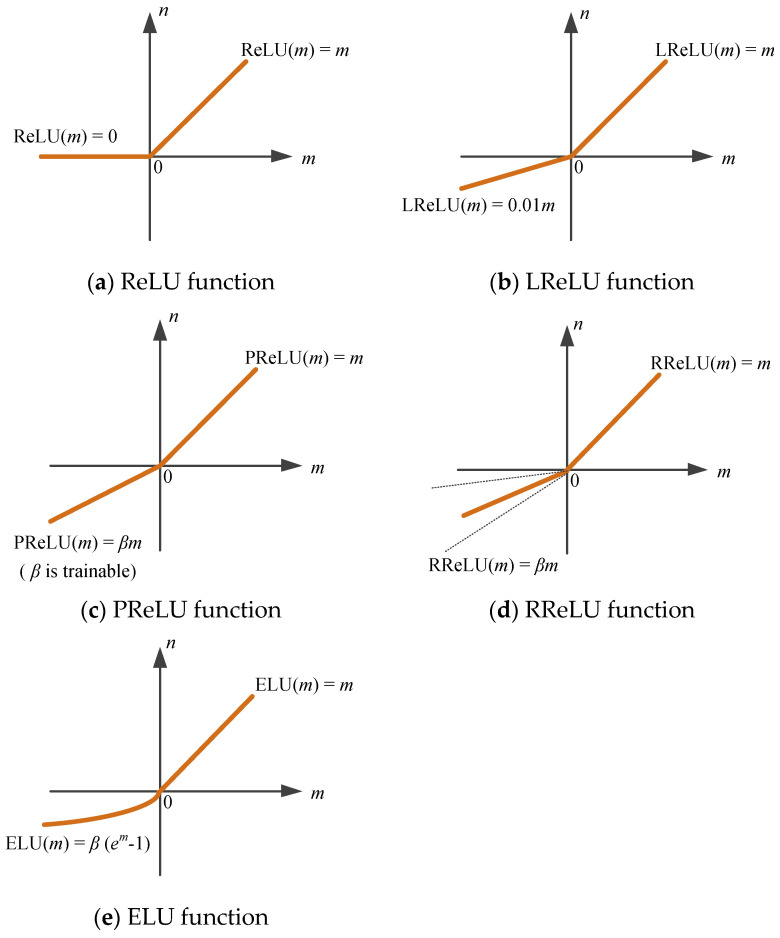
ReLU function and its variants.

**Figure 5 jimaging-09-00001-f005:**
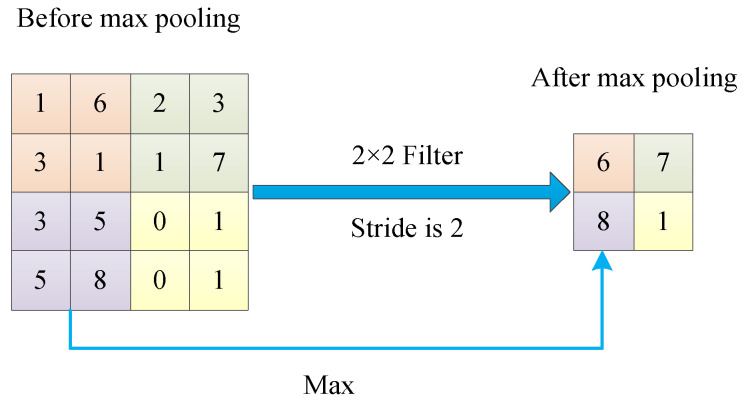
The process of max pooling.

**Figure 6 jimaging-09-00001-f006:**
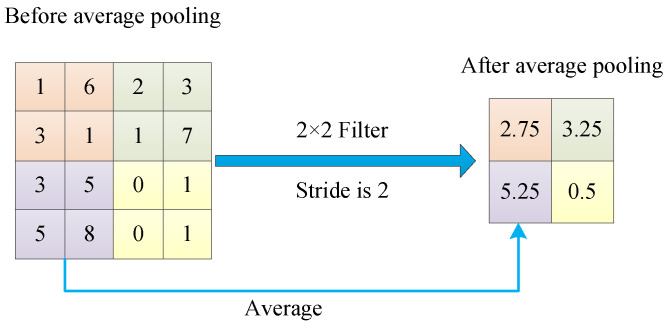
The process of average pooling.

**Figure 7 jimaging-09-00001-f007:**
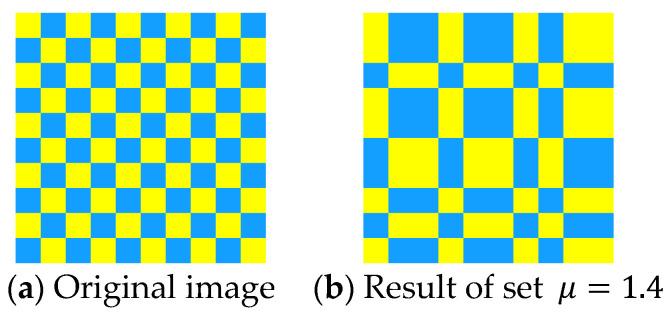
Illustration of FMP.

**Figure 8 jimaging-09-00001-f008:**
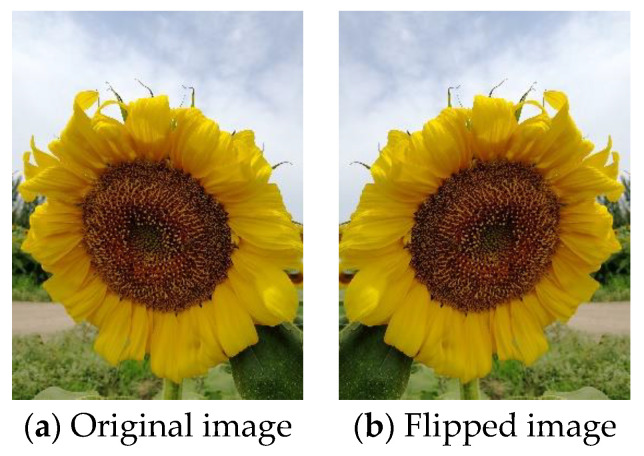
Horizontal flipping.

**Figure 9 jimaging-09-00001-f009:**
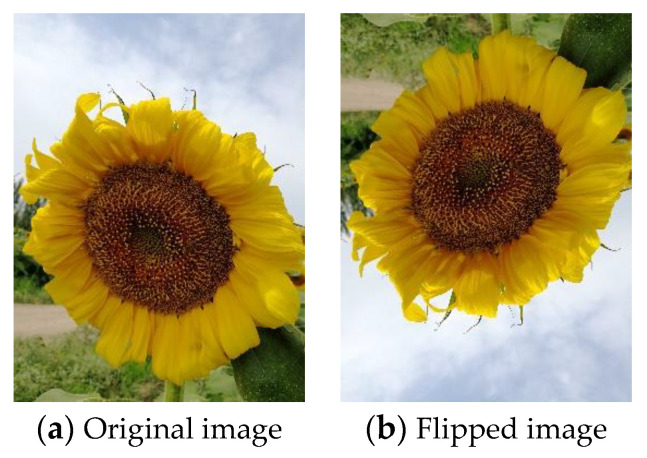
Vertical flipping.

**Figure 10 jimaging-09-00001-f010:**
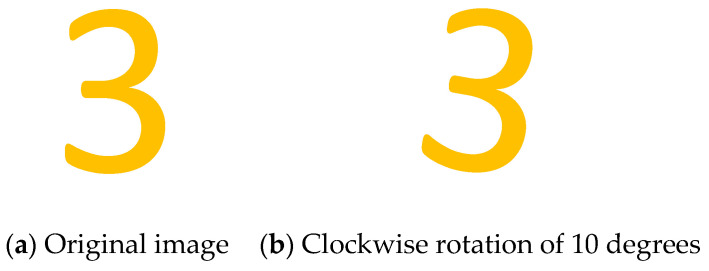
Rotation operation.

**Figure 11 jimaging-09-00001-f011:**
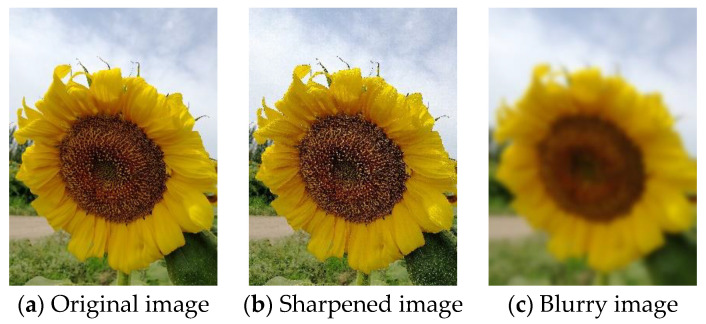
Sharpening and Blurring operation.

**Figure 12 jimaging-09-00001-f012:**
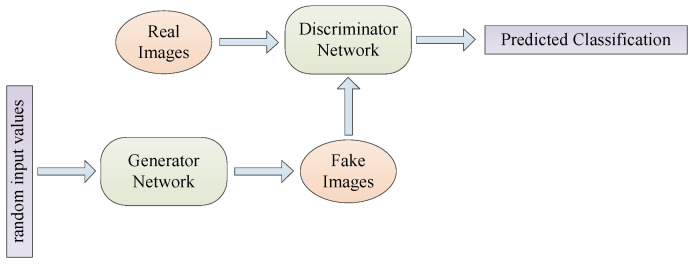
GAN architecture.

**Figure 13 jimaging-09-00001-f013:**
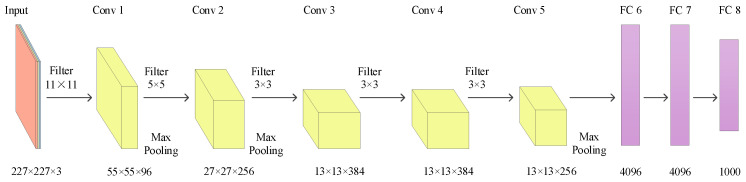
Structure of AlexNet.

**Figure 14 jimaging-09-00001-f014:**
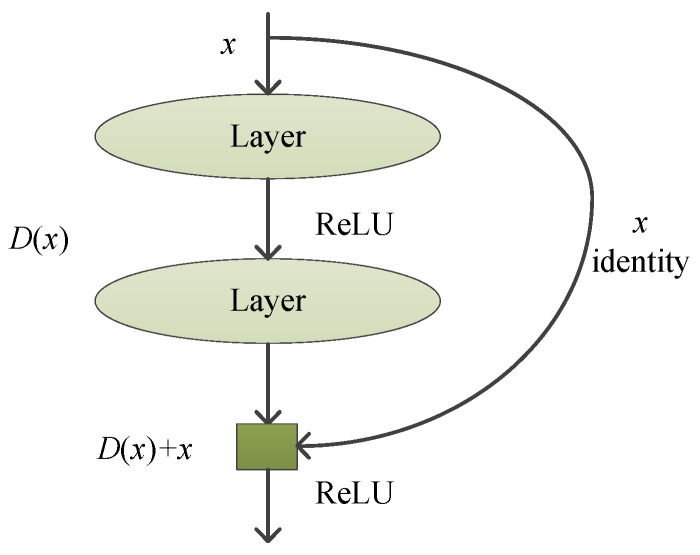
Shortcut Connection.

**Figure 15 jimaging-09-00001-f015:**
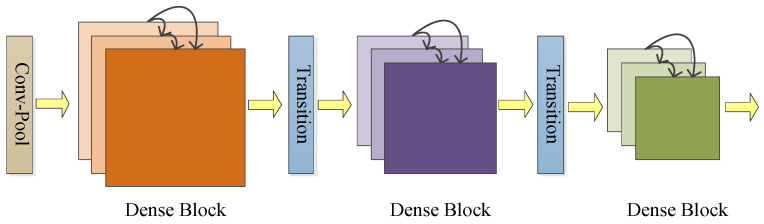
Structure of DenseNet.

**Figure 16 jimaging-09-00001-f016:**
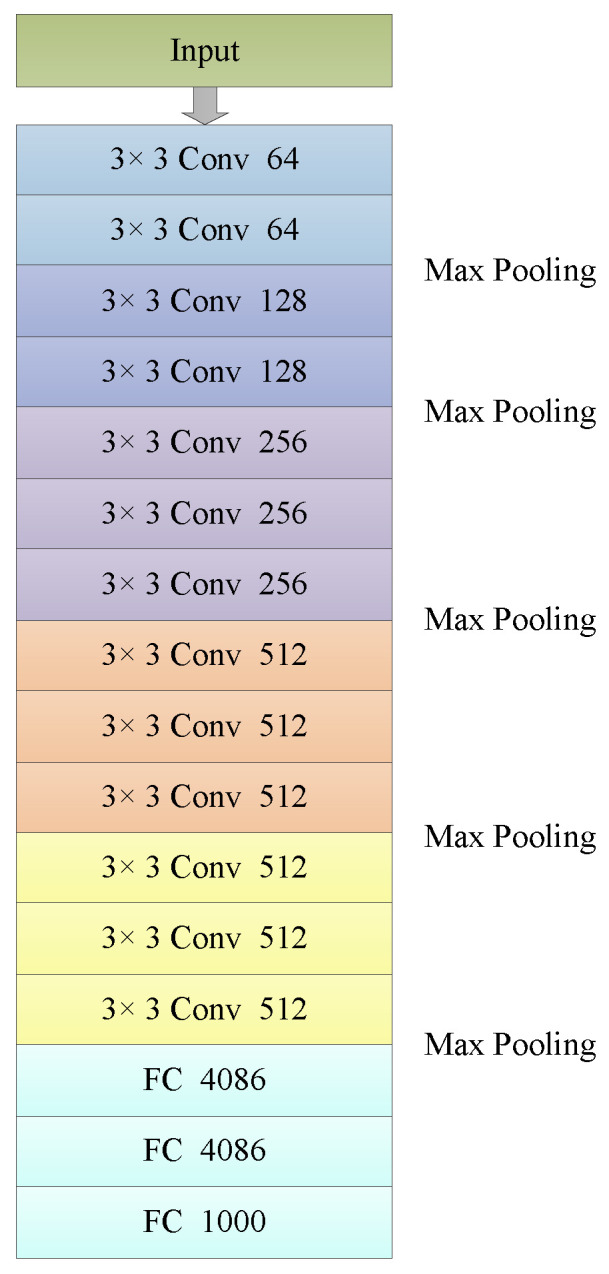
Structure of VGG16.

**Figure 17 jimaging-09-00001-f017:**
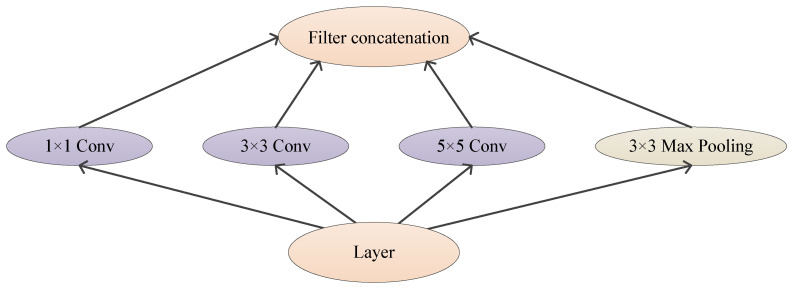
Basic Inception Module.

**Figure 18 jimaging-09-00001-f018:**
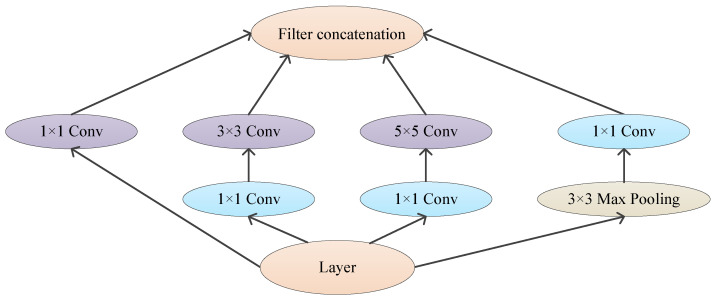
Inception module with 1×1 convolution.

**Figure 19 jimaging-09-00001-f019:**
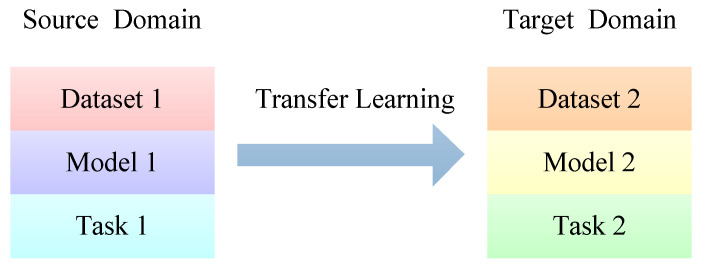
The process of transfer learning.

**Figure 20 jimaging-09-00001-f020:**
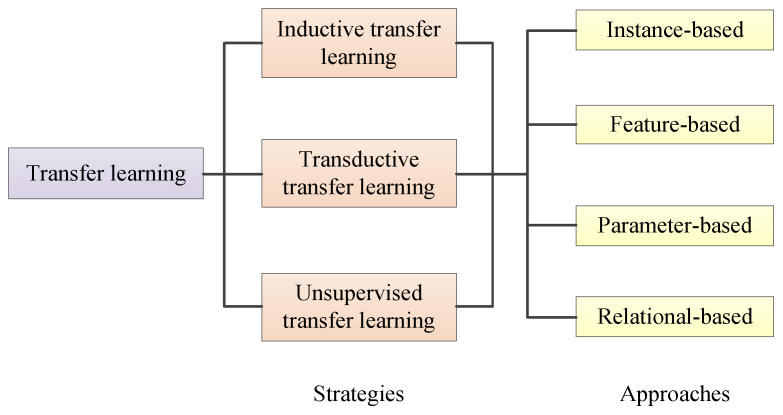
Strategies and approaches of transfer learning.

**Table 1 jimaging-09-00001-t001:** List of all abbreviations and terms.

Abbreviation	Term
AI	Artificial intelligence
ANN	Artificial neural network
AT	Angle transformation
AVNC	Attention-based VGG-style network for COVID-19
BiLSTM	Bidirectional long short-term memories
BN	Batch normalization
CB	Convolution block
CBAM	Convolutional block attention module
CGAN	Conditional generative adversarial net
CNN	Convolution neural network
CO-IRv2	Optimized inceptionResNetV2 for COVID-19
COVID-19	Coronavirus disease 2019
CT	Computed tomography
DA	Data augmentation
DC-Net	Deep COVID network
DCNN	Deep convolutional neural network
DenseNet	Dense convolutional network
DL	Deep learning
DNN	Deep neural network
EC	Electrochemical
ELU	Exponential linear unit
FCB	Fully connected block
FET	Field-effect transistor
FMP	Fractional max pooling
GANs	Generative adversarial networks
GGO	Ground glass opacity
GoogleNet	Google inception net
GPU	Graphics processing unit
ICS	Internal covariate shift
LReLU	Leaky rectified linear unit
LSTM	Long short-term memory
mAlexNet	Modified AlexNet
ML	Machine learning
PAM-DenseNet	DenseNet with parallel attention module
PDF	Probability density function
PReLU	Parametric rectified linear unit
ReLU	Rectified linear unit
ResNet	Residual network
RICAP	Random image cropping and patching
RReLU	Randomized leaky rectified linear unit
RT-PCR	Reverse transcription-polymerase chain reaction
RVFL	Random vector functional-link net
SE block	Squeeze and excitation blocks
SPEA-II	Strength Pareto evolutionary algorithm-II
SPR	Surface plasmon resonance
SRGAN	Super-resolution generative adversarial network
TSA	Tree seed algorithm
VGG	Visual geometry group
VSBN	VGG-style base network
WCNN4	Wavelet CNN-4

## Data Availability

Not applicable.

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
