# Peer review of "A Survey on Deep Learning in COVID-19 Diagnosis"

_2313-433X, 2022, doi:10.3390/jimaging9010001_

Round 1

Reviewer 1 Report

This paper proposes  A Survey on Deep Learning in COVID-19 Diagnosis. 

Following improvements can be performed before next submission

1. Authors have mentioned in the abstract "The convolutional neural network (CNN) is very popular now". This is not acceptable as it is a kind of vague statement. If you would like to mention then you need to mention on which context it is popular, and due to what features/properties it is popular.

2. Since there are many existing survey papers are exist, what is the newness or superiority of this survey?

3. Table 2 presents the results for various parameters of different technquies. My question here is whether all techniques used the same data set and control parameters.

4. This work mainly presents components of CNN or its parts. It has very less discussion on the impact of/on COVID-19.

5. What is the need to explain pooling, relu and all those things in the paper?

6. I would suggest the authors to discuss the implications of different deep learning techniques on COVID detection/identification.

Reviewer 2 Report

The manuscript focuses on reviewing the latest Deep Learning Convolutional Neural Networks systems developed for COVID-19 diagnosis both using chest X-ray and CT images. First, the authors describe what Convolutional Neural Networks are in general and not in the specific COVID-19 applications, analyzing in detail all the elements of these networks and the most used architectures. Then, they summarize the current works and methods developed in the literature for COVID-19 diagnosis using chest X-ray or CT medical images as starting data. The final objective claimed by the authors is to contribute to future research in this field. The research question is relevant to the state-of-the-art and a detailed review of the literature focused on Deep Learning in COVID-19 diagnosis is very useful given the current spread of research works in this field.  However, I have some concerns about the consistency of the present review article in the way it is presented. In particular, the following concerns should be addressed:

1)    The authors do not declare if they use a specific protocol for the systematic review, e.g., the PRISMA protocol. They do not specify what search criteria they adopted to find the cited works and select them, for example, if they use keywords in some specific academic database or archive (e.g., PubMed) and use specific filters, such as the year of publication or others.

2)    The manuscript appears more like a general review of Deep Learning and CNN in general than of Deep Learning techniques applied to COVID-19 diagnosis, as the title and the abstract suggest. Too much of the review and references are devoted to the description of the elements forming a CNN, which are described with too many details. CNNs applied to COVID-19, instead, are too briefly summarized in a too-brief paragraph so that it does not appear as the focus of the survey.

3)    A lot of references do not seem appropriate, as they do not contain links to what is said in the text. Moreover, there are too many references to works not focused on medical images.

4)    As the focus of the survey is on DL applied to COVID-19, a more precise description of the imaging modalities (CXR and CT) should be presented, including definitions for Ground glass and Consolidation (possibly with figures).

5)    The description of the CNN architectures is not entirely correct and clear. Some information does not apply in general.

6)    As already said, the survey of the literature on COVID-19 applications is far too summarized. In the various mentioned papers, the classification made by the CNN is not specified. What are the classes? COVID/Not COVID, Severe/Not Severe? I suggest the authors specify it, otherwise, the results of performances reported in the tables are not comparable and meaningful. I also suggest the authors add some considerations to the datasets used in the various mentioned works.

7)    In the tables, the lines are not aligned, and it is not clear what the references correspond to.

8)    I found the English too informal and unclear in certain passages.

Therefore, I suggest the authors consistently revise the manuscript and resubmit it after having addressed these issues.

Reviewer 3 Report

The authors did a great job summarizing many CNN papers on COVID-19 diagnosis. Here are some suggestions:

1. Reference 170 paper (Heidari et.al) discusses using a preprocessing algorithm on the images to improve accuracy. Similarly, if you find a few other papers that use other preprocessing techniques, stating all the preprocessing techniques would add more comprehensiveness to the paper.

2. In section 3 CNN: Please add the convolution formula. Given, n_in = number of input features, p = size of padding, k = kernel size, s = stride, the n_out(number of output features ) = floor((n_in+2p-k)/s)+1 

Round 2

Reviewer 1 Report

The authors have improved the survey work based on the given suggestions.

Reviewer 2 Report

I recognize the effort of the authors to improve the manuscript following the given suggestions. The main reported issues have been solved. In my opinion, the manuscript is now in acceptable form.